# Aortic Valve Defect as an Independent Risk Factor for Endothelial Dysfunction

**DOI:** 10.3390/cells14120885

**Published:** 2025-06-11

**Authors:** Mateusz Malina, Waldemar Banasiak, Adrian Doroszko

**Affiliations:** 1Clinical Department of Cardiology, Centre of Heart Diseases, 4th Military Hospital, 50-981 Wroclaw, Poland; 2Clinical Department of Cardiology, Faculty of Medicine, Wroclaw University of Science and Technology, 51-612 Wroclaw, Poland

**Keywords:** aortic stenosis, bicuspid aortic valve, endothelial function

## Abstract

Endothelial dysfunction (ED) has been identified as a precursor to micro- and macroangiopathic complications and an independent risk factor for major adverse cardiac events (MACEs). Recent studies have identified a novel risk factor for ED: severe aortic stenosis (AS). Traditionally linked to other established risk factors for endothelial cell dysregulation, AS has emerged as a contributor to ED, which is supported by the improvement of endothelial function following transcatheter (TAVR) or surgical (SAVR) interventions. Furthermore, the observation of ED in patients with a dysfunctional bicuspid aortic valve (BAV) at a younger age suggests a distinct impact of AS on ED. A promising hypothesis is a hemodynamic theory suggesting that changes in the shear stress of the ascending aortic wall and peripheral vessels, along with subclinical hemolysis caused by turbulent blood flow, could lead to reduced nitric oxide (NO) bioavailability. Current hypotheses on ED have yet to consider the influence of concomitant aortic stenosis in BAV. Additionally, studies examining potential intravascular hemolysis in BAV patients or the impact of surgical treatment of this defect on endothelial function are scarce. The aim of this review is to summarize the current knowledge on the mechanisms underlying ED in patients with AS or BAV and to identify possible directions for future research.

## 1. Introduction

Endothelial dysfunction (ED) is a broadly defined heterogeneous spectrum of pathophysiological phenomena that leads to the loss of endothelial homeostatic functions. This process is a recognized precursor of micro- and macroangiopathic complications [1,2] and an independent risk factor for major adverse cardiac events (MACEs) [3]. Numerous established and emerging risk factors contribute to endothelial dysregulation through various mechanisms, including hypertension, diabetes, obesity, dyslipidaemia, smoking, and other chronic diseases [4,5,6,7,8,9,10,11,12]. There are various methods to assess endothelial function; some allow for the evaluation of regional endothelial cell function, while others are used for systemic assessment. Invasive techniques are employed for regional functional assessment, involving evaluations of large vessels, such as coronary artery vasomotor responses measured by quantitative coronary angiography or intravascular ultrasound [13,14] as well as microcirculation assessments typically performed through coronary blood flow (CBF) measurements, with coronary flow reserve (CFR) considered the gold standard [15,16]. Conversely, non-invasive methods are widely used to evaluate systemic endothelial function [1,17,18], as they avoid the risks associated with invasive procedures. These include techniques such as the most widely used method, flow-mediated dilation (FMD) of the brachial artery [1,18,19], the laser Doppler flowmetry, and the peripheral arterial tonometry of arterioles located in the distal parts of upper extremities.

## 2. Aortic Stenosis (AS) as an Independent Risk Factor for Systemic Endothelial Dysfunction

Aortic valve disease manifesting as clinically significant stenosis has been identified as a potential independent risk factor for systemic endothelial dysfunction; however, this effect appears to be highly heterogeneous, as evidenced by studies showing reduced FMD values that do not consistently correspond with the severity of aortic stenosis [20]. There are several hypothetical mechanisms that explain the impact of AS on endothelial cell function. The potential etiopathogenesis of endothelial dysfunction in patients with AS is based on the reduced area available for blood flow due to the narrowed aortic valve. According to Bernoulli’s principle, this forces blood to flow through a smaller opening at a higher velocity [21]. Beyond the stenosis, the bloodstream disperses, causing irregularities in flow direction. These alterations generate turbulent blood flow, marked by the presence of vortices and velocity fluctuations, which result in localized increases in wall shear stress (WSS) within the ascending aorta. The high flow velocity generates greater frictional forces between the blood and the endothelium [22,23,24]. Concurrently, in peripheral arteries, decreased blood flow leads to reduced WSS [25]. The hemodynamic consequences in both the ascending aorta and peripheral arteries appear to influence ED. WSS is a critical biomechanical signal that regulates endothelial function. Proper WSS values promote vascular homeostasis by activating endothelial mechanoreceptors (e.g., integrins and glycocalyx complexes) and initiating signaling pathways that enhance nitric oxide (NO) production, suppress inflammation, and limit vascular smooth muscle cell (VSMC) proliferation [26,27]. Altered WSS values have been found to be strongly associated with the atherogenic process [28]. Low WSS in peripheral arteries has been shown to lead to the activation of pro-inflammatory factors, such as the nuclear factor kappa-light-chain-enhancer of activated B cells (NF-κB), resulting in an increased expression of inflammatory cytokines and adhesion molecules, like intracellular adhesion molecule 1 (ICAM-1) and vascular cell adhesion molecule 1 (VCAM-1). This, in turn, promotes leukocyte infiltration and the development of inflammation in the endothelium [29,30]. Additionally, endothelial nitric oxide synthase (eNOS) activity is reduced under low WSS conditions, leading to a decrease in NO bioavailability. NO bioavailability is a key factor for vasodilation and protection against atherogenic processes, such as platelet aggregation and VSMC proliferation [29]. Furthermore, gene expression profiles in endothelial cells change, predisposing them to a pro-atherogenic phenotype. These changes include increased vascular permeability, endothelial cell proliferation, and oxidative stress [29,31]. The disruption of the balance between protective and damaging processes is attributed to the insufficient stimulation of endothelial mechanoreceptors under low WSS [32,33,34]. The change in the metabolic profile of endothelial cells also occurs in response to increased shear stress [26]. Zeller J et al. proposed a hypothesis that increased WSS induces the accelerated dissociation of the pentameric isoform of the C-reactive protein (pCRP) into its monomeric subunits (mCRP) [35]. This form of CRP (mCRP) stimulates the secretion of pro-inflammatory cytokines, enhances leukocyte chemotaxis and adhesion to endothelial cells through the induction of adhesion molecule expression, and activates platelets by promoting their aggregation—thereby amplifying local inflammation [36,37,38]. The Piezo-type mechanosensitive ion channel component 1 (Piezo-1) may also be responsible for the direct activation of monocytes in response to increased wall shear stress [39]. The same receptor, along with other mechanoreceptors and vascular endothelial growth factor receptor 2 (VEGFR2), increases the expression of pro-inflammatory cytokines and induces phenotypic changes in valvular interstitial cells (VICs) under elevated wall shear stress, promoting calcification and thereby amplifying the mechanical stimulus. It is also possible that this vicious cycle mechanism applies to vascular endothelial cells as well, especially since endothelial cells usually exhibit a higher expression of VEGFR2 [40,41]. In the ascending aorta, increased WSS has been shown to have a detrimental effect on regional endothelial cells by activating signaling pathways such as NF-κB [29]. The injury to these endothelial cells may be reflected in elevated circulating endothelial microparticles (EMPs) [42]. Increased WSS may also disrupt the integrity of red blood cells, leading to the accumulation of extracellular hemoglobin (eHb) and hemoglobin microvesicles [43,44,45,46]. An indirect indication of increased hemolysis is the elevated level of the NO inhibitor, asymmetric dimethylarginine (ADMA), in individuals with significant AS [20], as red blood cells are potential reservoirs of ADMA [47]. The release of eHb and hemoglobin microvesicles during hemolysis leads to the conversion of NO to nitrites (NO2^−^) or nitrates (NO3^−^) through the action of NO scavengers. Furthermore, the release of arginase-1 (ARG-1) during hemolysis results in a reduction in arginine, the substrate for NO synthesis, thereby further diminishing the bioavailability of endogenous NO [48,49]. NO, in addition to its well-known vasodilatory function, has been shown to have protective effects against atherogenesis, including mitigating oxidative stress, platelet activation, inflammation, and VSMC proliferation [50]. Furthermore, hemoglobin microvesicles and free hemoglobin have been observed to participate in oxidative processes, generating reactive oxygen species (ROS) that accelerate inflammatory processes in the endothelium and damage its structure [51,52]. Furthermore, the release of ARG-1 during hemolysis reduces the availability of arginine, which is the substrate for NO synthesis [48,49]. Consequently, the bioavailability of endogenous NO is diminished. Nitric oxide, in addition to its well-known vasodilatory function, exerts protective effects against atherogenesis, including mitigating oxidative stress, platelet activation, inflammation, and smooth-muscle cell proliferation [50]. Furthermore, the previously mentioned arginase-1, in addition to decreasing the substrate for NO synthesis, promotes oxidative stress and is considered an important factor in the etiopathogenesis of ED in the presence of more established risk factors such as diabetes, hypertension, or obesity [53]. As a result of these phenomena, the systemic dysregulation of endothelial homeostasis occurs, which is reflected, among other things, by a decrease in FMD [10]. It has been posited that the circulating EMPs previously discussed do not contribute to the etiopathogenesis of systemic endothelial dysfunction. However, they are recognized as a marker of impaired endothelial integrity [54,55]. Furthermore, their levels have been observed to increase in the early stages of atherosclerosis [56], and they are acknowledged as negative prognostic markers of cardiovascular diseases [57,58] (Figure 1 and Table 1).

Aortic stenosis is the most common acquired valvular heart disease in developed countries, with the highest prevalence in the elderly population [59,60]. Thus, AS is frequently associated with other comorbid conditions such as hypertension, diabetes, obesity, and smoking [61,62,63]. These diseases are well-established risk factors for the development of endothelial dysfunction [64] and serve as major drivers in the progression of aortic valve sclerosis [65]. The coexistence of these ED-related risk factors with AS may serve as a counterargument to the hypotheses presented above. Opponents of the notion that AS is an independent risk factor for ED might argue that endothelial cell dysfunction in these patients arises from their comorbidities rather than from AS itself. Below, we present the existing body of scientific evidence and the conclusions drawn from it, which we believe lay the foundation for resolving this debate and recognizing AS to be an independent contributor to the development of ED. At the same time, we emphasize the necessity of further research to deepen our understanding of this issue.

### 2.1. Arguments Supporting the Notion of Aortic Stenosis as an Independent Risk Factor for the Loss of Endothelial Function

A compelling body of evidence substantiates AS to be an independent risk factor for ED, as evidenced by the enhancement of FMD after the management of significant aortic stenosis in patients exhibiting risk factors for endothelial dysfunction. Comella et al. [66] observed an augmentation in FMD following transcatheter aortic valve replacement (TAVR), both in the early and late follow-up phases, encompassing patients with risk factors for endothelial dysfunction (hypertension, type 2 diabetes, and smoking). A notable limitation of the study was the inclusion of a relatively small sample size (n = 27). In contrast, the study by Vitez et al. [67] demonstrated a significant enhancement in FMD in a group of 43 patients following TAVR in both early and late follow-up assessments. Additionally, an improvement in cardiac autonomic function, as measured by high-resolution ECG, was observed. The study by Moscarelli et al. [68] demonstrated a significant improvement in endothelial function in the late period after TAVR, and a significant improvement in FMD was also demonstrated after surgical aortic valve replacement (SAVR), with the improvement being greater in the surgical group compared to patients undergoing the transcatheter procedure.

Furthermore, Irace et al. [25] observed an increase in wall shear stress in the common carotid artery in patients undergoing SAVR, which could support the hypothetical decrease in WSS in the peripheral arteries as a potential component of endothelial dysfunction development. However, the impact on WSS in the ascending aorta, as well as other markers assessing endothelial cell function, were not examined.

In contrast, an earlier observation by Chenevard et al. [69] found no improvement in endothelial function in patients undergoing SAVR using the same ED marker as the Moscarelli et al. [68] group (FMD assessment).

Noteworthy, Horn et al. [70] observed an enhancement in FMD in patients following transcatheter aortic valve implantation (TAVI), concurrently detecting peripheral increases in WSS and a concomitant decrease in endothelial-derived microparticles. As previously delineated, EMPs are a recognized indicator of impaired endothelial integrity and a negative prognostic factor for cardiovascular diseases.

The most extensive study to date that has confirmed the improvement of endothelial function in patients with clinically significant AS undergoing TAVR was conducted by Quast et al. [71]. Utilizing patient observations, an experimental animal model with patient biological sample transfer, and computational fluid dynamics simulations, this study not only confirmed the improvement of FMD in a significant number of patients undergoing transcatheter aortic valve implantation, further solidifying the belief that AS is an independent risk factor for ED but also brought us closer to answering questions about the potential etiopathogenesis of this phenomenon. Observations during this study, such as the reduction in peak velocity in ascending aortic MR and a decrease in extracellular hemoglobin, erythrocyte microparticles, and hemoglobin microvesicles, suggest the possibility of a valid hypothesis regarding the phenomenon of WSS in the ascending aorta, causing subclinical hemolysis with subsequent consequences for systemic endothelial function. It is also noteworthy that even relatively low levels of extracellular hemoglobin, in the absence of overt signs of hemolysis, have been shown to reduce the bioavailability of endogenous NO, which is a finding that has also been confirmed in studies on stored red blood cells [72] (Table 2).

### 2.2. Bicuspid Aortic Valve (BAV) as Additional Potential Evidence of Independent Aortic Valve Dysfunction’s Impact on Impaired Endothelial Function

Another piece of evidence supporting the independent impact of significant AS on endothelial dysfunction may be its manifestation in the population of patients with a bicuspid aortic valve. These patients often develop significant aortic valve defects at a young age and are typically free from other recognized risk factors for ED [73,74]. While an established correlation exists between impaired endothelial function and BAV, as evidenced by various markers of endothelial dysregulation in patients with BAV, such as increased circulating EMPs [75] or decreased FMD levels [76,77,78], the number of theories about the development of ED in this population underscores the complexity of the underlying process, and the still unclear etiopathogenesis of this phenomenon, especially when associated with aortic dilation and/or aortic stenosis [75,77,79,80,81,82]. Genetic and hemodynamic theories have been identified as the predominant hypotheses. Recent studies have identified mutations in proteins such as Notch-1 [80], ROBO4 [83], or GATA4 [84] in patients with BAV. These proteins are considered key mediators in the signaling pathways involved in the dysregulation of the endothelial-to-mesenchymal transition (EndoMT) process. This process is regarded as crucial in heart development and wound healing [85]; thus, its dysregulation may be a direct cause of endothelial homeostasis loss [83]. Various microRNAs (miRNAs) also play a regulatory role in the proper progression of EndoMT. For instance, the microRNA-200 (miR-200) family appears to suppress this process [86]. While specific miRNA expression patterns have been associated with concurrent endothelial dysfunction in BAV, their precise role remains uncertain [87,88]. Another potential mutation is Knox20, which is responsible for the dysregulation of NO synthesis [89]. Hemodynamic hypotheses, however, largely focused on the local (in the ascending aorta) dysfunction of endothelial cells caused by increased WSS [90,91,92], also suggesting potential pathways leading to systemic dysfunction. A proper, laminar shear stress pattern leads to increased tension in platelet endothelial cell adhesion molecule 1 (PECAM-1), one of the most well-characterized mechanotransducers in endothelial cells (EC) [93,94], triggering its association with the vimentin cytoskeleton [95]. This interaction results in the activation of proto-oncogenic tyrosine kinase (Src), the vascular endothelial growth factor receptor (VEGFR), phosphatidylinositol 3-kinase (PI3K), and eNOS [96]. Furthermore, when the endothelium remains intact, other mechanosensitive pathways, such as transforming growth factor-beta (TGF-β), are activated. TGF-β plays a role in regulating the proliferation of vascular smooth muscle cells through endothelial heparan sulfate proteoglycans (HSPGs) [97,98] and leads to the activation of the Krüppel-like factor 2 (KLF2) signaling cascade [98,99]. KLF2 activation is responsible for inducing eNOS activity and production, thereby increasing NO synthesis, which is essential for maintaining vascular and endothelial tone [100]. In regions of turbulent blood flow, where shear stress is altered, the endothelial cell phenotype is modified, leading to increased endothelial permeability, the disruption of the proliferation-apoptosis balance, and enhanced monocyte adhesive properties [101]. What is more, the turbulent blood flow present in BAV leads to the impairment of the TGF-beta pathway, directly affecting phenotypic and structural changes in endothelial cells [98,99], reducing NO production [100], and dysregulating the previously mentioned EndoMT process [102,103]. Although reports indicate reduced eNOS expression in BAV patients [104], the observations made by Kotlarczyk et al. [105] provide an intriguing perspective. Their study identified regions of the ascending aorta with higher eNOS expression in BAV patients compared to those with TAV. However, despite this increased expression, they observed lower NO bioavailability. They hypothesized that this phenomenon results from oxidative stress induced by elevated WSS and an altered response to free radicals in smooth muscle cells (SMCs) of the medial layer of the ascending aorta in BAV patients [106,107]. Under oxidative stress conditions, including arginine deficiency and the depletion of the essential cofactor tetrahydrobiopterin (BH4) [108], eNOS can become “uncoupled” from NO production and instead generate the superoxide radical (O₂^●−^) [109]. Superoxide radicals oxidize tetrahydrobiopterin and react with NO, forming peroxynitrite (ONOO^−^), which further oxidizes tetrahydrobiopterin, perpetuating a vicious cycle of increasing oxidative stress, further eNOS uncoupling, reduced NO bioavailability, and generalized endothelial dysfunction [108,109]. It also suggests that turbulent blood flow may cause disruptions in the expression of proteins responsible for atherosclerosis processes, such as Calponin 1 (CNN1) [110] or Kruppel-like factor 4 (KLF4) [111] (Table 3).

Despite the evident association between the presence of BAV-AS and endothelial dysfunction, there is still a lack of evidence-based medicine (EBM) studies that quantitatively assess the relationship between the severity of AS in the context of BAV and endothelial dysfunction, for example, by evaluating the degree of impairment in flow-mediated dilation (FMD) or the levels of inflammatory and prothrombotic markers. There is currently a paucity of studies evaluating the potential presence of subclinical hemolysis, indicated by increased extracellular hemoglobin and hemoglobin microvesicles, and their potential impact on the bioavailability of endogenous NO in BAV patients. A potential indicator of increased hemolysis in patients with BAV could be the presence of elevated levels of ADMA, a well-documented NO synthase inhibitor, in this patient population [112]. This phenomenon has also been observed in patients with severely stenotic tricuspid aortic valves [20].

Additionally, there is a paucity of studies that directly compare endothelial function following BAV treatment with significant aortic defects. Conducting such research could facilitate a more comprehensive understanding of the pathogenesis of ED in patients with BAV and provide a framework for the appropriate qualification of patients with bicuspid aortic valves for surgical treatment.

## 3. Future Perspectives

The hypothesis that clinically significant aortic stenosis is an independent risk factor for systemic endothelial dysfunction has emerged recently, and its validity must be established through further research. Despite the growing number of studies and large body of evidence supporting the validity of this theory, the potential etiopathogenesis remains unclear. A promising avenue for furthering our understanding of the isolated impact of clinically significant AS on endothelial cell function lies in the study of patients with a bicuspid aortic valve, who typically do not have other recognized risk factors for ED development. The independent impact of BAV on the development of ED should be considered [78]; however, future studies are needed in order to analyze the potential role of genetic factors in the coexistence of BAV and ED. The impact of concurrent aortic valve defects on red blood cell integrity and vascular endothelial function should be taken into consideration, which could clarify the etiopathogenesis of ED in these patients (Figure 2).

## Figures and Tables

**Figure 1 cells-14-00885-f001:**
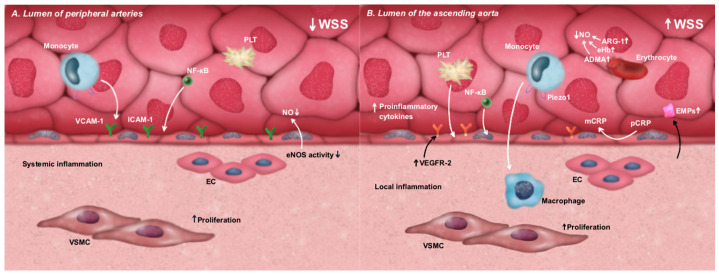
The impact of WSS disturbances on the metabolic shift in endothelial cells toward a pro-inflammatory phenotype and the associated ED. (**A**). Low WSS in peripheral arteries leads to the activation of the NF-κB pathway and the increased expression of adhesion molecules, such as ICAM-1 and VCAM-1, promoting leukocyte adhesion and transmigration. Simultaneously, low WSS reduces eNOS activity, resulting in decreased NO bioavailability, enhanced platelet aggregation, and VSMC proliferation. (**B**). High WSS in the ascending aorta activates mechanosensitive pathways involving NF-κB, VEGFR2, and Piezo-1 receptors, as well as the dissociation of pCRP into mCRP. These events contribute to the increased secretion of pro-inflammatory cytokines, leukocyte adhesion, platelet aggregation, and enhanced VSMC proliferation. Additionally, subclinical hemolysis—frequently present under high shear stress conditions—leads to the release of NO scavengers, further reducing NO bioavailability and exacerbating local inflammatory responses. ADMA—asymmetric dimethylarginine; ARG-1—arginase-1; EC—endothelial cells; ED—endothelial dysfunction; eHb—extracellular hemoglobin; EMPs—endothelial microparticles; eNOS—endothelial nitric oxide synthase; ICAM-1—intracellular adhesion molecule 1; mCRP—monomeric isoform of C-reactive protein; NF-κB—nuclear factor kappa-light-chain-enhancer of activated B cells; NO—nitric oxide; pCRP—pentameric isoform of C-reactive protein; Piezo-1—Piezo-type mechanosensitive ion channel component 1; VCAM-1—vascular cell adhesion molecule 1; VEGFR2—vascular endothelial growth factor receptor 2; VSMC—vascular smooth muscle cell; WSS—wall shear stress.

**Figure 2 cells-14-00885-f002:**
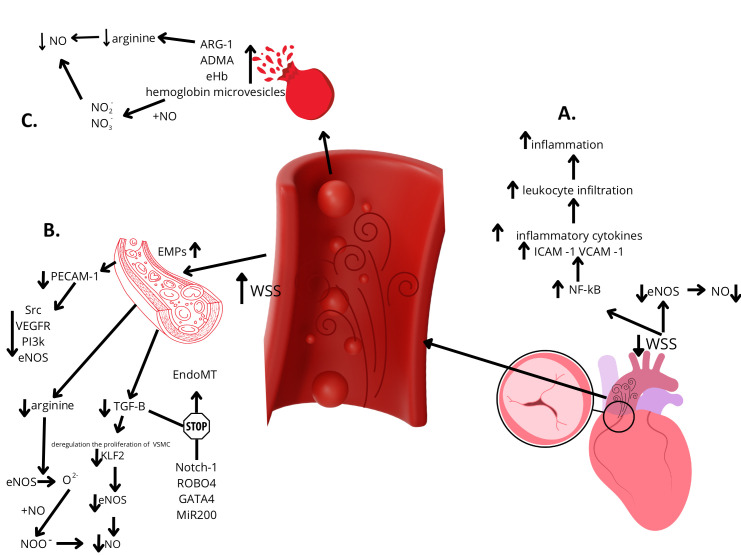
Hypotheses on the development of endothelial dysfunction in patients with bicuspid aortic valve stenosis. Legends: **A**—Lower shear stress within peripheral arteries leads to the decreased activity of eNOS and increased activation of the pro-inflammatory transcription factor NF-κB, resulting in enhanced local inflammation and reduced NO bioavailability. **B**—Increased shear stress within the ascending aorta leads to a reduction in the substrates necessary for nitric oxide (NO) synthesis, such as arginine, and impairs the activation of mechanosensitive pathways, including PECAM-1 and TGF-β. This results in decreased eNOS activity and reduced NO bioavailability. Furthermore, diminished TGF-β signaling, together with genetic factors, contributes to the inhibition of the EndoMT process. **C**—Subclinical hemolysis induced by localized increases in shear stress leads to the release of nitric oxide scavengers and arginine inhibitors—the key substrate for NO synthesis. This results in reduced nitric oxide bioavailability. ADMA—asymmetric dimethylarginine, ARG-1—arginase-1, EMPs—circulating endothelial microparticles; EndoMT—endothelial-to-mesenchymal transition; eHb—extracellular hemoglobin; eNOS—endothelial nitric oxide synthase; ICAM-1—intracellular adhesion molecule 1; KLF2—Krüppel-like factor 2; MiR200—microRNA-200; NF-kB—nuclear factor kappa-light-chain-enhancer of activated B cells; NO—nitric oxide; NO_2_^−^—nitrites; NO_3_^−^—nitrates; PECAM-1—platelet endothelial cell adhesion molecule 1; PI3k—phosphatidylinositol 3-kinase; Src—proto-oncogenic tyrosine kinase; TGF-β—transforming growth factor-beta; VCAM-1—vascular cell adhesion molecule 1; VEGFR—vascular endothelial growth factor receptor; VSMC—vascular smooth muscle cell; WSS—wall shear stress.

**Table 1 cells-14-00885-t001:** Causes of reduced nitric oxide (NO) bioavailability induced by changes in wall shear stress (WSS) and hemolysis in patients with clinically significant aortic stenosis (AS).

Potential Causes of Reduced NO Bioavailability in Patients with Severe AS
Reduced endothelial nitric oxide synthase activity due to decreased WSS in peripheral arteries
Increased activity of asymmetric dimethylarginine: an endogenous nitric oxide inhibitor
Conversion of NO to nitrites or nitrates due to extracellular hemoglobin and hemoglobin microvesicles
Upregulated arginase-1 activity, which is an enzyme responsible for the degradation of arginine, the primary substrate for nitric oxide synthesis

**Table 2 cells-14-00885-t002:** Experimental, clinical, and translational studies supporting individual hypotheses on the impact of aortic stenosis on endothelial dysfunction (the description of individual hypotheses in the pathophysiology of endothelial dysfunction development in the text).

	Experimental	Clinical	Translational
**Reduced wall shear stress (WSS) in peripheral arteries.**	D. C. Baeriswyl et al. [30]	C. Irace et al. [25]; A. S. Storch et al. [32];	
**Elevated wall shear stress (WSS) in the ascending aorta.**	P. van Ooij et al. [24]; A. M. Moerman et al. [28]; K. Katoh [29]; X. Liu et al. [41];	M. Michail et al. [23]; S. Baratchi et al. [39]; P. N. Diehl Ferenc et al. [42];	J. Zeller et al. [35]; S. U. Eisenhardt et al. [36]; J. R. Thiele et al. [38]; C. Quast et al. [71]
**Subclinical hemolysis.**	M. Frimat et al. [43]; S. Herold et al. [44]; R. F. Eich et al. [45]; A. Mahdi et al. [53]	R. P. Rother et al. [48]; C. R. Morris et al. [49]	C. Quast et al. [71]
**Increase in flow-mediated dilation after the applied intervention on the stenotic aortic valve.**		Comella et al. [66]; Vitez et al. [67]; M. Moscarelli et al. [68]; R. Chenevard et al. [69]; P. Horn et al. [70]	C. Quast et al. [71]

**Table 3 cells-14-00885-t003:** The classification of studies on the etiopathogenesis of endothelial dysfunction (ED) in patients with a bicuspid aortic valve (BAV) according to the investigated hypothesis: genetic, hemodynamic, or focusing on endothelial nitric oxide synthase (eNOS) activity or ED markers, taking into account both experimental and clinical studies.

	Experimental	Clinical
**Genetic theory**	S. Maleki et al. [83]; J. Gehlen et al. [84]; P. Poggio et al. [87]; N. Martínez-Micaelo et al. [88]; G. Odelin et al. [89]	C. R. Balistreri et al. [80];
**Hemodynamic theory**	C. Meierhofer et al. [91]; M. Osawa et al. [94]; D. E. Conway et al. [95]; I. Fleming et al. [96]; A. B. Baker et al. [98]; T. E. Walshe et al. [99]; R. D. Fontijn et al. [100]	N. Tzemos et al., 2010 * [76]; Y.-B. Wang et al. [78];
**Focused on eNOS or ED markers**	I. Fleming et al. [96]	N. Tzemos et al. [73]; J. M. Alegret et al. [74]; M. Vaturi et al. [79]; J. M. Alegret et al. [75]; D. Aicher et al. [104]; M. P. Kotlarczyk et al. [105]

*—The only study excluding patients with AS.

## Data Availability

No new data were created or analyzed in this study.

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
