# Peer review of "Aortic Valve Defect as an Independent Risk Factor for Endothelial Dysfunction"

_cells, 2025, doi:10.3390/cells14120885_

Round 1

Reviewer 1 Report

Comments and Suggestions for Authors

This is a very interesting manuscript by Malina and colleagues about the evidence of severe aortic stenosis (AS) as a novel risk factor for endothelial dysfunction (ED). The authors summarized the current hypotheses and knowledge on the mechanisms underlying ED in patients with AS or BAV and provided possible directions for future research. I have just a few minor suggestions that could improve the paper.

-Change in endothelial metabolic profile is considered an important contributing factor to ED. Is there any evidence or study linking aortic valve defect or AS to the alteration of endothelial metabolism?

-Page 1, Line 39, “coronary flow reserve (CRF)” should be CFR

-Abbreviations should be defined the first time they appear in the abstract; the main text; or the first figure/table. Some abbreviations are defined multiple times in the main text. Such as BAV, ED, FMD, WSS, EMPs, ARG-1, VSMC, etc.

Author Response

We would like to thank the Reviewer for an in-depth analysis of the manuscript and for pivotal comments provided, which have resulted in a significant improvement of this manuscript.

  1. Change in endothelial metabolic profile is considered an important contributing factor to ED. Is there any evidence or study linking aortic valve defect or AS to the alteration of endothelial metabolism?

Thank you for your interesting question and for providing an idea that helped us enrich our manuscript. In addition to the previously described effect of aortic stenosis (AS) on shifting the metabolic profile of endothelial cells toward a pro-inflammatory phenotype, we have taken the liberty of adding several additional hypotheses on this topic. The newly added section is shown below. We hope that these additions will make our manuscript an even more valuable source of knowledge in this field.

The change in the metabolic profile of endothelial cells also occurs in response to increased shear stress [35]. Zeller J et al. proposed a hypothesis that increased WSS induces the accelerated dissociation of the pentameric isoform of C-reactive protein (pCRP) into its monomeric subunits (mCRP) [36] This form of CRP (mCRP) stimulates the secretion of pro-inflammatory cytokines, enhances leukocyte chemotaxis and adhesion to endothelial cells through the induction of adhesion molecule expression, and activates platelets by promoting their aggregation—thereby amplifying local inflammation [37], [38], [39]. The mechanoreceptor Piezo-1 may also be responsible for the direct activation of monocytes in response to increased wall shear stress [40]. The same receptor, along with other mechanoreceptors and vascular endothelial growth factor receptor 2 (VEGFR2), increases the expression of pro-inflammatory cytokines and induces phenotypic changes in valvular interstitial cells (VICs) under elevated wall shear stress (WSS), promoting calcification and thereby amplifying the mechanical stimulus. It is also possible that this vicious cycle mechanism applies to vascular endothelial cells as well, especially since endothelial cells usually exhibit higher expression of VEGFR2 [41], [42].

  1. Page 1, Line 39, “coronary flow reserve (CRF)” should be CFR

We would like to thank the Reviewer for pointing out this issue. We have fixed it in the present version of the paper, according to the reviewer’s suggestion

  1. Abbreviationsshould be defined the first time they appear in the abstract; the main text; or the first figure/table. Some abbreviations are defined multiple times in the main text. Such as BAV, ED, FMD, WSS, EMPs, ARG-1, VSMC, etc.

Thank you for the valid comment. The definitions of abbreviations have been corrected throughout the text, as well as in the figures and tables. Additionally, we have included a list of all abbreviations along with their definitions at the end of the manuscript.

Reviewer 2 Report

Comments and Suggestions for Authors

This review lacks visual tools to better understand the major message. Please make several figures to the major aspects of the review. 'Picture 1', which I guess should be 'Figure 1', is too complex but at the same time very reductionist. Please split it is several major aspects. 

Author Response

We would like to thank the Reviewer for an in-depth analysis of the manuscript and for pivotal comments provided, which have resulted in a significant improvement of this manuscript.

  1. This review lacks visual tools to better understand the major message. Please make several figures to the major aspects of the review. 'Picture 1', which I guess should be 'Figure 1', is too complex but at the same time very reductionist. Please split it is several major aspects. 

We would like to thank the Reviewer for the suggestion regarding the graphical representation of the presented information. As the Reviewer suggested, we have added additional figures and changed the label from "Picture" to "Figure" for the previously included image. To facilitate a better understanding of the mentioned figure, we have also divided it into three sectors (A, B, C), which are described in the corresponding legend. We would like to retain this figure as the graphical abstract. We hope that in its current form, the manuscript is visually richer and more comprehensible.

Reviewer 3 Report

Comments and Suggestions for Authors

I very much enjoyed reading this review analyzing the potential role for aortic valve defect and endothelial dysfunction (ED). This is a very intriguing issue and the authors should be congratulated for their efforts. Though a large number of evidences have been resumed, here are a significant number of questions to be be clarified

In introduction there is some confusion between coronary and systemic ED: I guess that the authors should simply state that ED may be evaluated regionally or systemically, specifying different methods. 

As a general rule: what are we talking about? In particular, are we talking about aortic sclerosis (ASc) or stenosis (AS) ? 

Moreover, in ref 24 relates to sclerosis with V max < 2.5 m/s, and shows that its  is associated with systemic endothelial dysfunction. Noteworthy  reference 25 states that In patients with aortic stenosis increased flow-mediated dilation is independently associated with higher peak jet velocity with a non significant trend toward a higher average FMD (P = .12) in controls. This comparison raise the suspicion that the relationship between AS and FMD is not homogenous. 

Multiple studies, in experimental and clinical settings, have been produced on this topic. A table (with 2 separate fields, one for experimental, and the other for clinical settings) is needed to resume the quoted studies

Paragraph, as it is, is interesting, but confusing

BAV: are we talking about BAV per se of BAV-AS? Genetic background, pathophysiology of aortic dilatation with and without AS, are complicated: a Table here is needed. Stating “ Despite the numerous hypotheses concerning the etiopathogenesis of ED in patients with BAV, none of them accounts for the potential coexistence of significant aortic defects and assesses the degree of systemic ED depending on the severity of the coexisting defect.” After 60 lines regarding this topic, weakens the overall manuscript

Picture 1: there are (as I see) 3 “groups”. They should be labelled (A, B, C) and explained in a proper legend. 

Author Response

We would like to thank the Reviewer for an in-depth analysis of the manuscript and for pivotal comments provided, which have resulted in a significant improvement of this manuscript.

  1. In introduction there is some confusion between coronary and systemic ED: I guess that the authors should simply state that ED may be evaluated regionally or systemically, specifying different methods. 

We agree with the Reviewer, that the previous version of Introduction could be confusing. As result, we have stated that ED may be evaluated regionally or systemically, and the particular methods for its assessment have been specified as following:

„There are various methods to assess endothelial function; some allow evaluation of regional endothelial cell function, while others are used for systemic assessment. Invasive techniques are employed for regional functional assessment, involving evaluations of large vessels—such as coronary artery vasomotor responses measured by quantitative coronary angiography or intravascular ultrasound [13], [14]as well as microcirculation assessments typically performed through coronary blood flow (CBF) measurements, with coronary flow reserve (CFR) considered the gold standard [15], [16]. Conversely, non-invasive methods are widely used to evaluate systemic endothelial function [1], [17], [18], as they avoid the risks associated with invasive procedures. These include techniques such most widely used method—flow-mediated dilation (FMD) of the brachial artery [1], [18], [23], the laser doppler flowmetry and the peripheral arterial tonometry of arterioles located in the distal parts of upper extremities”

  1. As a general rule: what are we talking about? In particular, are we talking about aortic sclerosis (ASc) or stenosis (AS) ? Moreover, in ref 24 relates to sclerosis with V max < 2.5 m/s and shows that its  is associated with systemic endothelial dysfunction.

Thank you very much for your attentiveness and valid comment. This citation was included by mistake and has been removed in the current version of the manuscript.

Noteworthy reference 25 states that In patients with aortic stenosis increased flow-mediated dilation is independently associated with higher peak jet velocity with a non-significant trend toward a higher average FMD (P = .12) in controls. This comparison raises the suspicion that the relationship between AS and FMD is not homogenous. 

Thank you for the valuable observation. We agree that the previous version lacked emphasis on the heterogeneity of the impact of aortic stenosis (AS) on flow-mediated dilation (FMD). This has been addressed in the current version of the manuscript, and we hope it enhances the depth and detail of the discussion. In the revised version, the sentence now reads as follows:

Aortic valve disease manifesting as clinically significant stenosis has been identified as a potential independent risk factor for systemic endothelial dysfunction; however, this effect appears to be highly heterogeneous, as evidenced by studies showing reduced FMD values that do not consistently correspond with the severity of aortic stenosis.

  1. Multiple studies, in experimental and clinical settings, have been produced on this topic. A table (with 2 separate fields, one for experimental, and the other for clinical settings) is needed to resume the quoted studies

Thank you for this suggestion. We agree that such a table could help organize the studies and make it easier for the reader to navigate the topic we describe. An appropriate table, including three categories (experimental, clinical, and translational studies), has been included in the revised version of the review.

  1. BAV: are we talking about BAV per se of BAV-AS? Genetic background, pathophysiology of aortic dilatation with and without AS, are complicated: a Table here is needed.

Thank you for this comment. In response to the question, we are referring to BAV-AS as a more rapidly degenerating valve that generates shear stress disturbances in patients who have not yet developed clinically overt atherosclerosis or been exposed to the pressure of conventional risk factors for endothelial dysfunction. As recommended, we have also included a table, which we hope enhances the clarity of our manuscript.

  1. Stating “ Despite the numerous hypotheses concerning the etiopathogenesis of ED in patients with BAV, none of them accounts for the potential coexistence of significant aortic defects and assesses the degree of systemic ED depending on the severity of the coexisting defect.” After 60 lines regarding this topic, weakens the overall manuscript.

Thank you for this comment. We agree that the original sentence may have been insufficiently precise. What we intended to convey is the lack of studies quantitatively addressing the issue by assessing the severity of endothelial dysfunction in relation to the progression of aortic stenosis in patients with BAV. We hope that the revised sentence presented in the updated manuscript is clearer and more understandable:

Despite the evident association between the presence of BAV-AS and endothelial dysfunction, there is still a lack of evidence-based medicine studies that quantitatively assess the relationship between the severity of AS in the context of BAV and endothelial dysfunction, for example by evaluating the degree of impairment in flow-mediated dilation (FMD) or the levels of inflammatory and prothrombotic markers.

  1. Picture 1: there are (as I see) 3 “groups”. They should be labelled (A, B, C) and explained in a proper legend.

Thank you for rightly pointing out the unclear nature of the current figure. We agree with this assessment. In the revised version, in accordance with the recommendations, we divided the figure into three sections (A, B, C) and described them in the corresponding legend.

Round 2

Reviewer 3 Report

Comments and Suggestions for Authors

The authors answered exhaustively to my points

Author Response

We would  like to thank the Editor for all the comments provided. All the suggestions pointed out by the Editor have been addressed in the resubmitted version of the manuscript.